# Electrocardiographic Findings and Cardiac Troponin I Assay in Dogs with SIRS Diagnosis

**DOI:** 10.3390/vetsci9120655

**Published:** 2022-11-24

**Authors:** Michela Pugliese, Rocky La Maestra, Monica Ragusa, Mehmet Erman Or, Giordana Merola, Ettore Napoli, Annamaria Passantino

**Affiliations:** 1Department of Veterinary Sciences, University of Messina, Via Umberto Palatucci, 98168 Messina, Italy; 2Complex Structure of Surgical Sciences and Technologies, IRCCS—Scientific Institute for Research, Hospitalization and Healthcare—Istituto Ortopedico Rizzoli, Via Di Barbiano 1/10, 40136 Bologna, Italy; 3Faculty of Veterinary Medicine, İstanbul University-Cerrahpasa, Istanbul 34098, Turkey

**Keywords:** systemic inflammatory response syndrome (SIRS), cardiac dysfunction, myocardial damage, dog

## Abstract

**Simple Summary:**

Myocardial damage is commonly reported in humans during systemic inflammatory response syndrome (SIRS). Some authors hypothesize that it may also be present in dogs with SIRS myocardial damage; however, to date, this last finding is poorly investigated. The purpose of this study was to evaluate the potential cardiac involvement in dogs with SIRS. Clinical findings, cardiac troponin assay, and electrocardiographic alterations in two groups with different prognoses were analyzed and compared to a control group. The obtained results showed cardiac damage consequent to SIRS that influenced the prognosis.

**Abstract:**

Several studies performed in humans have demonstrated that the onset of systemic inflammatory response syndrome (SIRS) represents a high risk condition to develop myocardial damage and arrhythmias. Therefore, we also hypothesized cardiac involment for dogs affected by SIRS. To assess this hypothesis, 24 dogs with a diagnosis of SIRS (13 entire males, 7 entire females, and 4 spayed females) with an age ranging from 4 to 11 years (mean 5.6 years) and an average weight of 24 kg (range from 5 to 47 kg) were enrolled. The dogs were divided into two groups according to their prognosis: Survivors (G1) and not survivors (G2), composed by 13 and 11 dogs, respectively. Moreover, healthy dogs were included as the control group (CTR). All the dogs with a history of cardiac or renal disease were excluded. At the inclusion, each patient underwent a physical examination and a complete cell count, and a biochemistry panel (including electrolyte profile) was performed; moreover, the blood cardiac Troponin I (cTnI) was measured. For each clinical variable indicative of SIRS, a score between 0 (absence) and 1 (presence) was applied. Furthermore, an electrocardiographic examination was recorded. Seventeen out of 24 (70.8%) dogs with SIRS showed arrhythmias, of which n. 6 belonged to the G1, while n. 11 belonged to the G2. Most represented findings were sinus tachycardia (7/17; 41.1%), followed by monomorphic premature ventricular beats (6/17; 35.3%), less common were first-degree atrioventricular block (2/17; 11.7%) and sinus bradycardia 1/17; 5.8%). Notably, in G1 dogs, only sinus tachycardia and premature ventricular beats were observed. G2 dogs presented a number of total and banded leukocytes significantly higher than those of G1 (*p* = 0.002 and 0.049), in the same manner, the clinical score suggestive of SIRS (3 vs. 2.1) was significantly higher in G2 than in G1 dogs (*p* = 0.01). Moreover, a significantly higher value of cTnI was observed in the G2 group compared to the G1 group (*p* = 0.006). Data presented here suggested a cardiac involvement in dogs with SIRS, analogously to humans, that may significantly influence the patient’s prognosis.

## 1. Introduction

Systemic inflammatory response syndrome (SIRS) is a severe clinical condition characterized by an exaggerated defence response against different injurious agents that often result in organ dysfunction and death [1,2]. Sepsis [3], trauma [4,5], pancreatitis [6], and parasites [7,8] are the most common causes of SIRS reported for small animals. The mortality rate following SIRS is very high in dogs and cats, ranging between 20% and 68% [9]. SIRS can influence homeostasis and the functioning of various organs, such as the heart [10,11,12]. During SIRS, exaggerated inflammatory response causes a massive and unregulated release of nitric oxide and inhibits vascular α-adrenergic receptors, causing the relaxation of the vascular smooth muscle and a consequent systemic vasodilation that hesitate in a greater effort to maintain organ perfusion [13]. Moreover, as observed in humans, the release of high concentrations of IL-1β and TNF-α contributes to a decrease in cardiac function in patients affected by SIRS [13]. Patients with sepsis are at a greater risk of developing arrhythmias due to an altered myocardial excitability; therefore, the monitoring of cardiovascular function in these patients is considered to be a standard approach [14,15,16,17]. It is documented that patients with SIRS show organ failure and metabolic derangements, with a negative impact on the prognosis and survival [17]. Electrocardiographic abnormalities, especially arrhythmias, associated with an increase in cardiac Troponin I (cTnI), may be indicative of short- and mid-term prognosis, thus suggesting the possible onset of severe clinical sequelae [18]. The cTnI protein is one of three subunits of the troponin complex that has the function of regulating muscle contraction [19]. Over 90% of the cTnI of heart cells are bound to the contractile apparatus, while the remaining 10% is free in the cytosol [20]. When cardiac myocytes are damaged or lose membrane integrity, cTnI is released into the blood [20]. These characteristics make cTnI a biomarker of cardiac cell damage, both in human and in veterinary medicine [20,21].

The onset of cardiac dysfunctions during SIRS may be considered a priority, and if not treated in time, they can compromise the prognosis. Despite the relevance of SIRS in dogs and the high mortality rate correlated, the cardiac involvement in dogs affected by SIRS has been poorly investigated. In fact, as already observed in humans, it is reasonable to suspect, also in dogs, a cardiac involvement following SIRS [22,23,24,25,26]. The aim of the present study was to investigate the potential cardiac injury in dogs with SIRS having a different prognosis, through electrocardiographic examination and the assessment of cTnI blood levels. Data presented here provide relevant data for a better understanding of SIRS in dogs, improving the monitoring and prognosis related to SIRS management.

## 2. Materials and Methods

### 2.1. Ethics Statement

This study was conducted in accordance with standards reported by the Guide for the Care and Use of Laboratory Animals and Directive 2010/63/EU. The protocol and procedures employed were ethically reviewed and approved by the Ethical Committee of the Department of Veterinary Sciences of Messina University (No. 10/2018). Animals were only enrolled in the study after the signature of proper informed written consent by the owner.

### 2.2. Study Group

Twenty-four dogs with SIRS diagnosis were included in the study. The criteria for SIRS diagnosis were based on the finding of two or more of the following clinical abnormalities (i.e., body temperature < 37.7 °C or >39.7 °C; pulse rate > 160 beats per minutes (bpm); respiratory rate > 40 breaths per minutes (brpm); white blood cells > 2000/mm^3^ or <4000/mm^3^; banded leukocyte > 10%) [3]. Dogs weighing < 5 kg, aged < 6 months or >12 years, with primary cardiac diseases or with kidney diseases [27] were excluded.

Upon arrival at the admission, all the dogs underwent an assessment of basic clinical variables, including blood pressure measurement (VET HDO; S and B MedVet, Babenhausen, Germany). Medical history was also collected. A scoring system to categorize the clinical findings was applied. Body temperature, pulse rate, respiratory rate, white blood cells and banded leukocytes were considered as variables. Each variable was scored from 0 to 1 according to the alteration of the clinical sign considered (Table 1).

The total score of each patient was calculated as the sum of the score given to the single variable. Based on the survival from SIRS, dogs were divided into two groups: Survivors (G1 = 13 dogs) and not survivors (G2 = 11 dogs).

A control group (CTR) was constituted of n.10 adult mixed-breed dogs. They were considered healthy, based on the physical examination, haematological and biochemistry evaluations, electrocardiogram, and echocardiographic examination.

### 2.3. Electrocardiographic Examination

In each enrolled dog, standard electrocardiographic examination within 8 h admission was performed using 12-channel electrocardiography (Delta Tre Plus, Cardioline, Milano, Italy). Briefly, dogs were placed in the right lateral recumbency, with flattened electrodes attached to the skin over the olecranon on the caudal aspect of the forelimb, and over the patellar ligaments on the cranial aspect of the hindlimbs for bipolar limb leads and unipolar augmented limb leads [28]. Precordial leads were recorded by positioning precordial lead V_1_ at the costochondral junction of the right first intercostal space [29], while all left-sided leads (V_2_ to V_6_) were placed at the sixth intercostal space [30].

Isopropyl alcohol for a better connection between electrodes and the skin was applied [30]. Electrocardiogram (ECG) traces were set to 50 mm/s paper speed and 1 cm = 1 mV for 5 min. The evaluation of ECGs was performed according to the standard methods [31]. ECG variables, including the RR interval, PQ interval, QRS duration, and QT interval were calculated for all the leads [32].

QRS duration was calculated from its onset to the ST segment. PR interval was calculated from the start of P wave to the onset of QRS complex. QT interval was measured from the start of the QRS complex to the end of the T wave. An average of 5 cardiac cycles were defined as a result [32]. The sinus rhythm for twenty beats on lead II was determined. The Bailey hex-axial method to determine the mean electrical axis (MEA) in the frontal plane was applied [31,32]. Arrhythmias were classified according to the origin and duration [28].

### 2.4. Collection of Blood Samples and Laboratory Procedures

Blood samples, about 5 mL, were collected from the jugular or cephalic vein and were divided into two tubes. A tube with K_3_EDTA to perform a complete cell blood count (CBC) and to determine the concentration of plasmatic cTnI, while a tube without additives was used for biochemistry and ions evaluations. Alanine aminotransferase, alkaline phosphatase, glucose, urea, total proteins, albumins, creatinine, sodium (Na^+^), potassium (K^+^), chlorine (Cl^−^), and total calcium (Ca^2+^) were assayed. The cTnI levels were evaluated immediately using the VIDAS high sensitivity Troponin I Ultra assay (BioMérieux, Marcy-l’Étoile, France), in accordance with the manufacturer’s instructions (i.e., lower limit cTnI = 0.02 ng/mL; cTnI levels plasma range = <0.03–0.07 ng/mL) [33,34].

In detail, 150 μL of the sample was collected and transferred to the wells containing the anti-cTnI antibodies, determining the formation of a precipitate into the solid phase at the bottom. The presence of cTnI was therefore highlighted by the conjugated enzyme present on the bottom (4-Methyl-umbelliferone). This hydrolyzing formed a fluorescence directly proportional to the amount of cTnI present in the serum sample under examination. Every single run had an average of 17 min.

### 2.5. Statistical Analysis

A statistical analysis was performed using SPSS software (version 17.0, SPSS, Inc., Chicago, IL, USA). Normality was assessed by the Shapiro-Wilk test. Laboratory and electrocardiographic variables were expressed as descriptive variables and were analyzed as quantitative. Data showing normal distribution (clinical and electrocardiographic variables) were described as mean ± standard deviation (SD). Non-normally distributed data (cTnI) were described as the median and interquartile range (IQR, 25th and 75th percentiles).

To compare the parametric data, a t-student test was used, while a Mann-Whitney *U*-test for non-normally distributed non-parametric data was applied. Differences were considered significant for a *p*-value ≤ 0.05.

## 3. Results

All dogs of G1 and G2 showed two or more clinical findings suggestive of SIRS. Of the 24 dogs affected, 13 were males, 11 were females (4 spayed) and belonged to different breeds (n. 16 crossbreeds, n.4 German Shepherds, n.2 Rottweiler, n.1 Jack Russell Terrier and n.1 Beagle). The mean age was 5.6 years (range from 4 to 11 years), and the mean body weight was 24 kg (range from 5 to 47 kg) (Table 2). No significant differences in CBC and biochemical tests (data not shown) were recorded.

The mean onset of clinical signs prior to admission was 24 ± 11 h. No treatments were performed prior to admission. In the enrolled dogs, the SIRS origin was due to gastroenteritis n.6 (25%), pneumonia n.5 (20.84%), closed cervix pyometra n.4 (16.6%), trauma in n.4 cases (16.6%), pancreatitis n.3 (12.5%), and the presence of an intestinal foreign body n.2 (8.3%) (Table 3).

Data on body temperature, pulse rate (bpm), respiratory rate (arm), and total and banded leukocytes of G1, G2, and CTR groups are summarized in Table 4. Significant differences in body temperature were observed between G2 and CTR (*p* = 0.003). In addition, significant differences between G1 and CTR were detected in the pulse rate (*p* = 0.003), respiratory rate, the number of total leukocytes (*p* = 0.004) and band neutrophils (*p* = 0.001).

In G2, the total number of leukocytes and the percentage of band neutrophils were significantly higher than the G1 Group (*p* = 0.002 and 0.049, respectively).

Body temperature was altered in six dogs of G1 (46.2%) and in four (36.4%) dogs of G2, respectively. The pulse rate was increased in three (23.1%) dogs of G1 and in four (36.4%) of G2, while the alteration of the respiratory rate was detected in five (38.5%) dogs of G1 and three (27.3%) of G2. All dogs included in G1 and G2 showed alteration in the number of leukocytes. Leukopenia was recorded in five (38.5%) dogs of G1 and seven (63.6%) dogs of G2, respectively. The SIRS clinical score of G2 was was significantly higher (3 ± 0) than those of G1 dogs (2.1 ± 0.33) (*p* = 0.01). G1 and G2 showed values of clinical score and cTnI significantly higher than CTR (*p* = 0.01 and *p* < 0.01, respectively). The median value of cTnI in G1 and G2 dogs was 26.1 (IQR 9.4–89) and 3301 (IQR 229.1–3530.8), respectively, revealing a significantly higher value in G2 (*p* = 0.006) (Table 5). Values of mean blood arterial pressure were lower in dogs in G1 (95.34 ± 19.5) and G2 (92.11 ± 16.4) than CTR (106.2 ± 12; G1 vs. CTR, *p* = 0.04; G1 vs. CTR, *p* = 0.03).

Quantitative data of the ECG variables are summarized in Table 6. All dogs included in CTR presented electrocardiographic variables into normal ranges. No significant differences in ECG variables were observed between G1 and G2, while statistical difference (*p* < 0.05) for most of the ECG variables between G1 and CTR, and G2 and CTR (Table 6) was detected. 

Seventeen dogs (70.8%) showed electrocardiographic abnormalities, six (out of 13 46.1%) belonging to the G1, and eleven (100%) belonging to G2. The most observed findings were sinus tachycardia (7/17; 41.1%), followed by monomorphic premature ventricular beats (6/17; 35.3%), less common are first-degree atrioventricular block (2/17; 11.7%), and sinus bradycardia (1/17; 5.8%). The heart rate mean during sinus tachycardia in G1 was 175.5 ± 19, while in G2 it was 179.4 ± 17.8. No significant statistical differences were detected. The mean heart rate in dogs with first-degree atrioventricular block was 100 ± 10, while the dog with bradycardia had a heart rate of 52 bpm.

Regarding the type of arrhythmias found in dogs with SIRS, and in the G1 group only sinus tachycardia and monomorphic premature ventricular beats were observed, while other alterations together, with the latter were observed in the G2 (Figure 1). The interval between sinus beats and premature ventricular beats was 0.44 ± 0.2 with the right (n.4) and left bundle morphology (n.2).

## 4. Discussion

The present study demonstrates the cardiac involvement in dogs affected by SIRS and the relevance of the cTnI value as a predictive parameter for the prognosis. As stated elsewhere [3,4,5,6] the first cause of SIRS in dogs are infectious diseases followed by inflammatory status, for instance pancreatitis. According to the results, the prognosis of patients affected with SIRS may be predicted by monitoring several clinical variables, such as body temperature, which was significantly higher in dogs with a worse prognosis. Conversely, the leucocytes count, and the percentage of banded leucocytes were less indicative. In dogs, the measurement of cTnI is used as a biomarker for the damage of the myocardium, even if similarly, to humans, considerable increases can be observed in several severe diseases, such as gastric dilatation-torsion [35], infections and trauma [21,36,37]. The exact mechanism that leads to the increase of cTnI in SIRS affected dogs is not clear and still debated [24,36,37,38]. The pathogenesis of inflammation-inducing cardiac dysfunction during SIRS is not understood, even if it has been suggested that the generalized inflammatory hyperreactivity triggered by infectious factors and non-infectious host stimulatory agents are involved [22,39]. Among the pathological alterations that lead to cardiac contractile dysfunction and to a cTnI increase in SIRS patient are myocardial necrosis, hypertension and decreased renal clearance. The onset of cardiac dysfunction notes as “myocardial hibernation”, secondary to SIRS, has been reported in humans and experimentally in dogs [12,40,41]. Moreover, myocardial hibernation has been correlated with an increase in blood concentrations of cTnI, and with a poor prognosis [18,42,43]. The majority of examined dogs showed increased values of cTnI, however higher blood levels were found in G2 dogs. Therefore, cTnI measurement should be considered as an indicator of a life-threatening condition and related to poor prognosis, and consequently patients should have intensive medical care.

Dogs with a poor prognosis (G2) had a SIRS’s clinical score higher than G1. In fact, the dogs that died during hospitalization (G2) showed a mean SIRS clinical score of three, while dogs with a favorable prognosis (G1) scored 2.1. Therefore, the presence of more than two clinical signs recognized as criteria for SIRS diagnosis may be related to a worse prognosis. The ECGs findings underlined the presence of arrhythmias in both the groups, however dogs in G1 presented exclusively sinus tachycardia and premature ventricular beats, while animals in G2 presented several arrhythmias. While the causes of arrhythmias in the course of SIRS are still debated [14,15,16,17,44], even in humans, several studies report the presence of alterations in ECG tracing such as sinus tachycardia [17,44], atrial fibrillation [16,45], decrease in the amplitude of the QRS complex [14] and alterations of the QT interval [15] in critically ill patients. The mechanism related to the onset of these abnormalities is not completely understood, but it seems that an altered myocardial excitability consequent to a state of distress in the cardiovascular system is present. In fact, SIRS determines an increase of the blood requirement in peripheral tissues due to vascular hypotension which is added to cardiac depression [12]. The absence of electrolyte abnormalities suggests the presence of other mechanisms responsible for arrhythmias in SIRS, as the possible onset of myocardial hypoxia [44]. Furthermore, abnormal activation of the sympathetic nervous system may occur during sepsis, which could play a key role in the genesis of tachyarrhythmias [45].

## 5. Conclusions

Cardiac damage is considered to be a main complication in patients with several viral or bacterial infections, and trauma associated with SIRS. An important limitation of the study was not having the post-mortem cardiac histopathology in order to better define cardiac injuries during SIRS in dogs. Other limitations is the small number of animals included and the lack of an echocardiographic study. Further studies are necessary to better clarify this topic.

It is interesting to underline that dogs with a greater number of clinical findings are suggestive of SIRS, and therefore have a higher score and worse prognosis.

The cardiac involvement in dogs with SIRS markedly influences the prognosis and the cTnI should be considered to be a relevant negative prognostic marker, especially when it is associated with more than two clinical diagnostic criteria and electrocardiographic abnormalities.

The cTnI should be considered to be a relevant parameter for the formulation of the prognosis in dogs with SIRS diagnosis, especially when coupled with the clinical and electrocardiographic findings.

## Figures and Tables

**Figure 1 vetsci-09-00655-f001:**
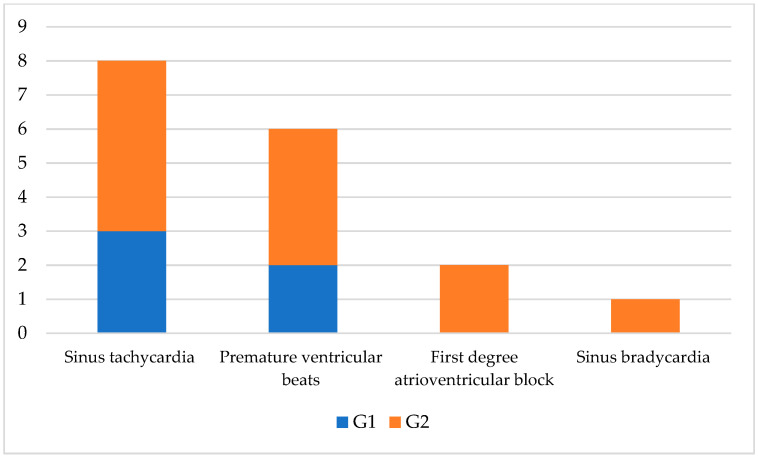
Graphical representation of arrhythmias detected in ECGs of SIRS affected dogs with a favorable (G1) and poor (G2) prognosis.

**Table 1 vetsci-09-00655-t001:** A scoring system used for SIRS diagnosis in dogs enrolled.

Variables	0 (Normal Range)	1 (Out of Normal Range)
Body temperature	>37.7 °C or <39.7 °C	<37.7 °C or >39.7 °C
Pulse rate	<160 bpm	>160 bpm
Respiratory rate	<40 brpm	>40 brpm
White blood cells	<12,000/mm^3^ or >4000/mm^3^	>12,000/mm^3^ or <4000/mm^3^
Banded leukocyte	<10%	>10

bpm = beats per minute; brpm = breaths per minutes.

**Table 2 vetsci-09-00655-t002:** Signalment of dogs included in G1 and G2.

	G1 (n.)	G2 (n.)
Male	6	8
Female	7	3
Crossbreed	10	6
German Shepherd	3	1
Rottweiler	0	2
Jack Russel	0	1
Beagle	0	1

**Table 3 vetsci-09-00655-t003:** Origin of SIRS in dogs belonged G1 and G2.

	G1 (n.)	G2 (n.)
Gastroenteritis	4	2
Pneumonia	4	1
Closed cervix pyometra	2	2
Trauma	1	3
Pancreatitis	1	2
Intestinal foreign body	1	1

**Table 4 vetsci-09-00655-t004:** Mean and standard deviation of clinical variables in dogs with a favorable (G1) and poor (G2) prognosis and CTR.

Clinical Variables	G1	G2	CTR
Temperature (°C)	38.2 ± 1.78	39.0 ± 1.82 ^b^	38.0 ± 1.24 ^b^
Pulse rate (bpm)	130 ± 14.1 ^a^	153.3 ± 72.1 ^b^	90 ± 15 ^ab^
Respiratory rate (brpm)	37 ± 9.9 ^a^	40.7 ± 14.5 ^b^	20.1 ± 4.2 ^ab^
Leukocytes (10^3^/μL)	10.53 ± 1.8 ^a^	17.44 ± 8.84 ^b^	8.20 ± 3.7 ^ab^
Band neutrophils (%)	6.6 ± 3.5 ^a^	9.1 ± 4.6 ^b^	2.1 ± 0.6 ^ab^

^ab^ Lowercase letter in the superscripts indicate a statistical significance between rows (*p* < 0.05).

**Table 5 vetsci-09-00655-t005:** Comparison of SIRS scoring and cTnI value between dogs with a favorable prognosis (G1), poor prognosis (G2) and control (CTR).

Variables	G1	G2	CTR	Normal Values [21]
Clinical Score	2.2 ± 0.43 ^ab^	3 ±0.7 ^ab^	0.9 ±0.3 ^ab^	-
cTnI (ng/mL)	88.2 ± 151 ^a^	1794.7 ± 1853 ^b^	0.10 ± 0.02 ^ab^	0.00–0.11

^ab^ Lowercase letter in the superscripts indicate a statistical significance between rows (*p* < 0.05).

**Table 6 vetsci-09-00655-t006:** Comparison of electrocardiographic parameters between animals with a favorable prognosis (G1), poor (G2) prognosis and control (CTR). Data are expressed as the mean and standard deviation.

Electrocardiographic Variables	G1	G2	CTR
Heart rate (bpm)	148.3 ± 72.2 ^a^	188 ± 122.1 ^b^	88.2 ± 12.8 ^ab^
MEA (degree)	57.8 ± 38.7	64.6 ± 13.7 ^b^	56.1 ± 14 ^b^
P duration (ms)	40 ± 0 ^a^	32 ± 17.8	34.8 ± 0.22 ^a^
P amplitude (mV)	0.14 ± 0.05 ^a^	0.12 ± 0.08 ^b^	0.22 ± 0.05 ^ab^
PR duration (ms)	73.3 ± 10 ^a^	88 ± 76.9 ^b^	101 ± 10.1 ^ab^
QRS duration (ms)	40 ± 0	45 ± 12.9 ^b^	40.8 ± 2.6 ^b^
R amplitude (mV)	0.83 ± 0.57	1.0 ± 0.2 ^b^	0.74 ± 0.41 ^b^
ST deviation (mV)	0	0.01 ± 0.003	0
T amplitude (mV)	0.25 ± 0.26	0.32 ± 0.38 ^b^	0.20 ± 0.17 ^b^
QT duration (ms)	176 ± 60.6 ^a^	228 ± 50.4 ^b^	194 ± 22 ^ab^
Corrected QT (ms)	178 ± 55 ^a^	230 ± 48	215 ± 19 ^a^

^ab^ Lowercase letter in the superscripts indicate a statistical significance between rows (*p* < 0.05).

## Data Availability

Data may be found, contacting the corresponding author (michela.pugliese@unime.it).

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
