# Peer review of "Electrocardiographic Findings and Cardiac Troponin I Assay in Dogs with SIRS Diagnosis"

_vetsci, 2022, doi:10.3390/vetsci9120655_

Round 1

Reviewer 1 Report (Previous Reviewer 1)

Dear Authors,

I find your article to be very informative and pertinent to veterinary medicine. You have a sound study with clear results. 

Unfortunately the manuscript still needs English language revisions. It is difficult to follow the text and I feel I'm reading a rough draft, not a final version. You have put in considerable efforts , however there is still many corrections to be made. A native speaker would be beneficial to improve manuscript, although I've have included many suggestions in the first part of the text, which I have attached. 

Author Response

Dear Reviewer,

We would like to thank for her/his time and for all the valuable suggestions.

Herein you find a revised version the manuscript according to your comments.

The changes made in the manuscript to address comments are written in red.

  1. Dear Authors,

I find your article to be very informative and pertinent to veterinary medicine. You have a sound study with clear results. 

Unfortunately the manuscript still needs English language revisions. It is difficult to follow the text and I feel I'm reading a rough draft, not a final version. You have put in considerable efforts , however there is still many corrections to be made. A native speaker would be beneficial to improve manuscript, although I've have included many suggestions in the first part of the text, which I have attached. 

Unfortunately, we did not find your file attached with your suggestion. However, English language revisions were performed to improve the quality of the manuscript.

Kind Regards

Prof. Annamaria Passantino

Prof. Ettore Napoli

Reviewer 2 Report (New Reviewer)

Comments on the attached manuscript.

Author Response

Dear Reviewer,

We would like to thank the reviewer for her/his time and for all the valuable suggestions.

Herein you find a revised version the manuscript according to the reviewers’ comments.

The changes made in the manuscript to address comments are written in red.

  1. Lines 47, and 48 of the introduction, "Systemic inflammatory response syndrome (SIRS) is a severe clinical conditioncharacterized by an unregulated release of inflammatory cytokines", the authors depended only on the leukocytes as indication of inflammatory responseand  did not measure cytokines. Why did not you measure the cytokines indicative of SIRS?
  2. We are grateful for the suggestion, given that may be considered an important deepening for furthers studies, but the measurement of cytokines is not into the aims of this study. The sentence has been rephrased (line 53-55).
  3. Line 100, the authors mentioned that dogs with kidney diseases were excluded, please interpret the cause in the discussion section.
  4. Dogs affected by kidney diseases may present an increase of cTnI, therefore we have decided to apply that as exclusion criteria. The appropriate reference has been included in the text (Sharkey LC, Berzina I, Ferasin L, Tobias AH, Lulich JP, Hegstad-Davies RL. Evaluation of serum cardiac troponin I concentration in dogs with renal failure. J Am Vet Med Assoc. 2009 Mar 15;234(6):767-70) (line 107-108).
  5. The word disfunction in lines 19, 43, and 276 should be corrected to dysfunction.
  6. Done
  7. In table 1, please confirm that the abbreviation "arm"  is correct or should be modified to "apm".
  8. The footnote has been corrected.
  9. In tables 2, 3, and 4, thesignificance letters are missed on some data, please check
  10. Done
  11. In table 4, since the study comprised a control group, there is no need to mention the normal ranges.
  12. Done

Kind Regards

Prof. Annamaria Passantino

Prof. Ettore Napoli

Reviewer 3 Report (New Reviewer)

In the present study, the authors tried to investigate the cardiac involvement in dogs affected  with SIRS. The following points are important to consider:

Major concerns:

1. Lines 47, and 48 of the introduction, "Systemic inflammatory response syndrome (SIRS) is a severe clinical condition characterized by an unregulated release of inflammatory cytokines", the authors depended only on the leukocytes as indication of inflammatory response  and  did not measure cytokines. Why did not you measure the cytokines indicative of SIRS?

2. Line 100, the authors mentioned that dogs with kidney diseases were excluded, please interpret the cause in the discussion section.

Minor concerns:

1. The word disfunction in lines 19, 43, and 276 should be corrected to dysfunction.

2. In table 1, please confirm that the abbreviation "arm"  is correct or should be modified to "apm".

3. In tables 2, 3, and 4, the significance letters are missed on some data, please check.

4. In table 4, since the study comprised a control group, there is no need to mention the normal ranges.

Author Response

Dear Reviewer,

We would like to thank for her/his time and for all the valuable suggestions.

Herein you find a revised version the manuscript according to your comments.

The changes made in the manuscript to address comments are written in red.

- Lines 47, and 48 of the introduction, "Systemic inflammatory response syndrome (SIRS) is a severe clinical conditioncharacterized by an unregulated release of inflammatory cytokines", the authors depended only on the leukocytes as indication of inflammatory responseand  did not measure cytokines. Why did not you measure the cytokines indicative of SIRS?

We are grateful for the suggestion, given that may be considered an important deepening for furthers studies, but the measurement of cytokines is not into the aims of this study. The sentence has been rephrased (lines 47-49).

- Line 100, the authors mentioned that dogs with kidney diseases were excluded, please interpret the cause in the discussion section.

Dogs affected by kidney diseases may present an increase of cTnI, therefore we have decided to apply that as exclusion criteria. The appropriate reference has been included in the text (Sharkey LC, Berzina I, Ferasin L, Tobias AH, Lulich JP, Hegstad-Davies RL. Evaluation of serum cardiac troponin I concentration in dogs with renal failure. J Am Vet Med Assoc. 2009 Mar 15;234(6):767-70) (line 101).

- The word disfunction in lines 19, 43, and 276 should be corrected to dysfunction.

Done

- In table 1, please confirm that the abbreviation "arm"  is correct or should be modified to "apm".

The footnote has been corrected.

- In tables 2, 3, and 4, thesignificance letters are missed on some data, please check

Done

- In table 4, since the study comprised a control group, there is no need to mention the normal ranges.

Done

Kind Regards

Prof. Annamaria Passantino

Prof. Ettore Napoli

Reviewer 4 Report (New Reviewer)

The title in not adequate. Cardiac involvement it is not only ECG.

Lack of the number of dogs with scores according to the SIRS scoring system.

Lack of count HRV, P wave dispersion, corrected QT.

Lack of statistical analysis ECG parameters.

Lack of information about blood presure.

Lack of HR during sinus tachycardia, bradycardia, AV block.

Laco of interval between sinus beat and VPc. 

Lack of information about morfology VPs.

Why echocardiographic parameters have not been assessed?

The statement that "The cTnI index should be considered an important parameter in formulating prognosis" is generally known in cardiology

Author Response

Dear Reviewer,

We would like to thank for her/his time and for all the valuable suggestions.

Herein you find a revised version the manuscript according to your comments.

The changes made in the manuscript to address comments are written in red.

- The title in not adequate. Cardiac involvement it is not only ECG.

The title has been changed in according the suggestion.

- Lack of the number of dogs with scores according to the SIRS scoring system.

Included.

- The dogs with SIRS diagnosis are 24, They are

Yes, they are.

- Lack of count HRV, P wave dispersion, corrected QT.

We did not calculate HRV and P wave dispersion, given that unfortunately included in the study design. QT correction has been applied.

- Lack of statistical analysis ECG parameters.

The statistical analysis of ECG parameters is yet included in the text (lines 219-246).

- Lack of information about blood presure.

The measurement of blood pressure was performed. The values have been included in the text (Line 103-105; 209-211).

- Lack of HR during sinus tachycardia, bradycardia, AV block.

The required information has been added in the revised version of the manuscript (lines 233-234).

- Laco of interval between sinus beat and VPc. 

The required information has been added in the revised version of the manuscript (lines 234-237).

- Lack of information about morfology VPs.

The required information has been added in the revised version of the manuscript (lines 234-237).

- Why echocardiographic parameters have not been assessed?

The echocardiographic evaluation is not into the aims of this study, considering electrocardiographic findings and cTnI assay.

- The statement that "The cTnI index should be considered an important parameter in formulating prognosis" is generally known in cardiology.

The sentence has been rephrased (lines 313-317).

Kind Regards

Prof. Annamaria Passantino

Prof. Ettore Napoli

Round 2

Reviewer 2 Report (New Reviewer)

The manuscript has been dramatically improved. 

Author Response

Reviewer 3 Report (New Reviewer)

The authors have revised the manuscript well. The overall quality of the manuscript has been improved. They addressed all issues i concerned. I recommend publication in the present form.

Author Response

Reviewer 4 Report (New Reviewer)

In my opinion, this study without echocardiogarphy results should not be published.

Author Response

This manuscript is a resubmission of an earlier submission. The following is a list of the peer review reports and author responses from that submission.

Round 1

Reviewer 1 Report

The authors presented their work entitle Cardiac involvement in dogs with the SIRS diagnosis. The topic is presently relevant in clinical practice and the work is interesting and presented in a precise manner.

As is, the article can be understood, but there are significant language revisions needed throughout the whole work and this would be best with the help of a native speaker.

The citations need a bit of revision as well as there are several positions that are included twice, ie. 13 and 43, 21 and 44, as well as 10 and 46. Also the citations are properly annotated in the text up to citation number 38, 39 is omitted and everything from number 40 onwards is not in order. Finally the article cites a work number 49, however this is not in the list of References. Otherwise the References section is appropriate.

The research presented is scientifically sound, however I would make a suggestion related to materials and methods (if possible results) and the discussion section. First of all there is no control group. I think that it would be fairly easy to gather a group of healthy animals and include them as a third, control group.  I also suggest that the authors include information about the ions in the blood work analysis. This should have been a part of the routing blood work-up in ailing dogs and because the concentration of ions has an influence on the heart rhythm, this information would be a useful addition. I believe the results would not change the outcome of the groups, but rather could shed a bit of light on the reason (or lack thereof) for the presents of more severe arrhythmias in the G2 dogs. In addition the topic of measuring blood gases should be touched upon at least in the Discussion section. This is not necessarily a routine test, but might be warranted in future works on myocardial damage. Finally, a comment should be made about breeds that are prone to arrhythmias as this might change the outcome of a study directed at arrhytmia analysis. In your study groups you had ie. German Shepherds (maybe a minor predisposition to rhythm abnormalities) and Rottweiler (possible DCM breed predisposed to arrhythmias).

The Tables and Figure are clear and appropriate and nicely present the study groups and the results. Table 3 should give the units for cTnI. The statistical analysis is appropriate. The ethics statement is adequate.

Some minor corrections include:

line 24 - should read mean 5.6 years

line 50 - the heart is part of the cardiovascular system, consider rephrasing   

line 52 - NO is an abbreviation that should be explained when used for the first time, also check for other abbreviations without explanations

line 211 - the "A" in the middle of the sentence should be lower case

 I think this work is sound and need some revisions described above before publication.

Reviewer 2 Report

We would like to thank the authors for their interesting manuscript and acknowledge the incredible amount of work that was put into this manuscript.  Unfortunately, there is, in my opinion, a major flaw in the study design – the lack of an echocardiogram.  The argument is that the troponin elevation is indicative of myocardial involvement due to SIRS however this cannot be substantiated based on the information provided.  The fact is an echocardiogram needs to be performed at baseline (i.e. while in the hospital) to assess cardiac function (and frankly ideally show LV dilation and systolic dysfunction) and then once the patient has recovered showed resolution of the systolic dysfunction on repeat echocardiogram at least 10-14 days later. (This also rules out concurrent structural cardiac disease as a contributing factor to the troponin elevation rather than it being from SIRS)  This is the current definition in human medicine as well as the criteria for previously reported case reports (Dickinson AE, et al JVIM 2007; 21(5): 1117-20).  As such it is frankly impossible to state if the troponin elevation is due to underlying SIRS vs underlying cardiac disease and therefore impossible to prove ‘cardiac involvement in SIRS’.  The authors can state that the ECG is normal however frankly an ECG is incredibly insensitive at ruling in or ruling out structural cardiac disease in dogs.  

I believe there is a typographical error in Table 1 as both say > 10% for banded leukocyte

We do not have any information regarding what is the cause of the SIRS, (it says infectious disease in 16/24? But does not apparently include pneumonia (n = 1), pyometra (2) which would be considered infectious diseases (right/) So what kind of infectious diseases are we talking about? 

What was the timing of the testing? i.e. within 8 hours of hospitalization? Within 24 hours of hospitalization? This should all be standardized and I’m not sure it is. 

Who is interpreting the ECGs? No offense if there was a cardiologist available my assumption is this person would have done the echocardiograms so how are you accounting for the fact that a cardiologist may/will have a different interpretation from a clinician with a different specialty?  How can you therefore definitively state an ECG is ‘normal’? 

Why is a sinus tachycardia (7/17) or sinus bradycardia (1/17) considered an ‘arrhythmia’ this would not be considered an arrhythmia.  Also what is a ‘junctional rhythm”? Like a junctional escape beat? A junctional tachycardia? This all goes back to the who is interpreting these findings and without a cardiologist it is difficult to say these results are publishable (I’m sorry I know it’s harsh but it is true).  

Again I did not address the conclusion because again (respectfully) I do not believe the conclusions of the study can be justified by the data or study design.